# Dietary Betaine Impacts Metabolic Responses to Moderate Heat Exposure in Sheep

**DOI:** 10.3390/ani13101691

**Published:** 2023-05-19

**Authors:** Kristy DiGiacomo, Sarah Simpson, Brian J. Leury, Frank R. Dunshea

**Affiliations:** 1School of Agriculture, Food and Ecosystem Science, Faculty of Science, The University of Melbourne, Parkville, VIC 3010, Australiabrianjl@unimelb.edu.au (B.J.L.); fdunshea@unimelb.edu.au (F.R.D.); 2Faculty of Biological Sciences, The University of Leeds, Leeds LS2 9JT, UK

**Keywords:** dietary supplement, metabolism, metabolic challenges, physiology, ruminant

## Abstract

**Simple Summary:**

Heat stress elicits metabolic and physiological responses in ruminants as they attempt to maintain thermal balance. Dietary betaine is an osmolyte that can reduce physiological responses to heat (such as core temperature). In this experiment, we examined the effects of dietary betaine supplementation and mild heat stress in response to metabolic (glucose, insulin, and adrenocorticotropic hormone) challenges in sheep. Our novel results indicate that the positive effects elicited by betaine supplementation are in part due to betaine influencing adipose tissue metabolism and the actions of insulin.

**Abstract:**

Dietary betaine supplementation can ameliorate physiological responses to heat exposure (HE) in sheep. This experiment measured metabolic responses to glucose (intravenous glucose tolerance, IVGTT), insulin (insulin tolerance test, ITT), and adrenocorticotropic hormone (ACTH) challenges in Merino ewes (*n* = 36, 39.7 kg) maintained at thermoneutral (TN, 21 °C) or HE (18–43 °C) and supplemented with either 0, 2, or 4 g/day dietary betaine (*n* = 6 per group). Sheep had ad libitum access to water and were pair-fed such that the intake of the TN sheep mimicked that of the HE sheep. After 21 days of treatment, sheep were fitted with jugular catheters and subjected to consecutive daily challenges (IVGTT, ITT, and ACTH, d 21–23, respectively), followed by skeletal muscle and subcutaneous adipose tissue biopsy collections for gene expression analysis (d 24). The HE-treated sheep had a greater insulin:glucose ratio (*p* = 0.033), a greater estimated homeostatic model assessment of insulin resistance (HOMAIR; *p* = 0.029), and a reduced revised quantitative insulin sensitivity check index (RQUICKI; *p* = 0.015). Sheep fed betaine (2 + 4 g/day) had a greater basal plasma insulin (*p* = 0.017) and a reduced basal non-esterified fatty acid (NEFA; *p* = 0.036) concentration, while the RQUICKI was reduced (*p* = 0.001) in sheep fed betaine. The results suggested that betaine supplementation alters lipid metabolism by potentially improving insulin signaling, although these responses differ between TN and HE conditions. There was no other impact of temperature or dietary treatments on the tissue gene expressions measured. Our results support the notion that betaine, in part, acts to modify lipid metabolism.

## 1. Introduction

When production animals are exposed to high ambient temperatures (HE), for example during summer heat waves, they can incur negative impacts on efficiency and welfare. These negative impacts are often preceded by a cascade of responses triggered by HE, ultimately influencing water balance [1], cellular physiology and metabolism, and gene expression pathways [2]. Production animals regularly experience HE due to the combined effects of increased ambient temperatures and the high metabolic rates observed in modern high-producing breeds. As ruminants produce a high volume of heat during the fermentation and digestion of feed, they are particularly susceptible to such negative responses [3]. However, sheep can maintain a relatively stable core temperature in response to HE [4], compared to larger ruminants. Sheep employ various methods to dissipate heat, including increased respiration rates (RR) and redistribution of blood flow towards the periphery [2]. If heat dissipation pathways are insufficient, physiological changes such as increased core temperature and changes in the secretion of hormones such as prolactin will occur [5]. Metabolically, HE also increases circulating insulin concentrations in ruminants, and in lactating ruminants, HE increases circulating and glucose-stimulated insulin secretion while adipose tissue reserves are not mobilized even when animals go into negative energy balance [6,7,8,9]. In response to heat and other stressors (such as oxidative stress, prolonged ischemia, and exposure to stress hormones such as adrenocorticotropic hormone (ACTH), corticosterone, and epinephrine [10,11]), cells rapidly accumulate heat shock proteins (HSPs). These HSPs perform three main biochemical functions: acting as chaperones, regulating cell redox balance, and regulating protein turnover [12]. Individual tissue types (adipose and skeletal muscle) have inherently different baseline temperatures [13], and thus the expression of tissue HSPs likely varies between tissue types. Chauhan et al. [14] found that the mRNA of heat shock transcription factor 1 (HSF1), a highly conserved mediator of the heat shock response, increased in response to heat in crossbred sheep. Heat shock protein 70 (HSP70) is a key HSP that is the most readily induced and has been shown to increase in plasma from heat-stressed growing feedlot cattle [15] and in the skeletal muscle of heat-stressed crossbred sheep [14].

Maintaining production animal performance under HE requires a combination of processes, including environmental modifications (e.g., shade or sprinklers) [5], feed additives and dietary manipulations [16], or genetic selection for heat-tolerant breeds [17,18]. Dietary manipulation and dietary supplements are often the easiest methods to implement to ameliorate the negative effects of HE in animal production systems and can address the underlying metabolic and cellular changes that result from HE. One such supplement is betaine (trimethylglycine), which acts as an organic osmolyte or methyl donor. Betaine is energy sparing [19] by reducing the energy required to maintain cellular osmotic balance via ion pumps [20] and by promoting cell proliferation and preventing apoptosis [21]. Betaine also impacts adipose tissue metabolism and has been shown to decrease fat depths in growing lambs [22] but increase fat depths in more mature cattle [23,24]. In pigs, betaine increased circulating somatotropin (GH), insulin-like growth factor 1 (IGF1), and insulin concentrations [25], although there is no evidence of this occurring in ruminants. In addition, betaine increases the melting temperature of macromolecules while also increasing the survival of heat-stressed bacteria via methods that are believed to involve HSPs [26]. We recently demonstrated that 2 g/day of dietary betaine reduced physiological markers of heat stress (including RR and rectal temperature), while 4 g/day of betaine increased these parameters in HE sheep. We also observed that both 2 and 4 g/day of betaine decreased plasma non-esterified fatty acid (NEFA) concentrations (~25 μM) [27], consistent with results observed in lactating cattle [28]. Understanding the metabolic responses to different doses of dietary betaine will improve producers’ ability to tailor supplementation regimes to specific situations. The aim of this experiment was to investigate the metabolic responses and changes to gene expression in adipose and skeletal muscle tissues to two doses of dietary betaine (2 and 4 g/day) fed to sheep maintained under HE or thermoneutral (TN) conditions. We hypothesize that, similar to the physiological responses observed [27], dietary betaine supplementation will ameliorate the negative effect of HE on insulin production and increase tissue HSP mRNA expression in sheep.

## 2. Materials and Methods

### 2.1. Animals, Housing, and Treatments

Procedures for these studies were approved by the University of Melbourne’s Faculty of Veterinary and Agricultural Sciences Animal Ethics Committee (ID 1011620.2). The animals and feed used in this experiment are as described previously [27]. Briefly, 36 female 8–10-month-old Merino ewes (39.7 ± 3.1 kg; 2–3 cm fleece length) were selected from the same flock and acclimatized to the experimental diets and facilities over 14 days prior to the 3-week experiment. Sheep were allocated across three replicates into one of six possible treatment groups (*n* = 6 per treatment): 0 (control), 2, or 4 g betaine/day (betaine 96% feed grade, FeedWorks Pty Ltd., Romsey, Victoria, Australia), exposed to either high environmental temperature (HE) or TN conditions. The recommended dose of betaine in pigs is 0.125% of intake, and in this experiment we selected increased doses (0.16 and 0.30% of feed intake, respectively) to account for likely degradation by rumen microbes [29]. Betaine was introduced into the diets at half the experimental concentration over the final 3 days of the acclimatization period. Sheep were fed a diet consisting of 70% alfalfa hay and 30% oat chaff (approximately 10 MJ of ME, CP 13%), split over twice daily feedings (0900 and 1600 h). Betaine supplements were offered at the morning feeding prior to the forage portion to ensure the full dose was consumed. As HE typically reduced feed intake, sheep were pair fed, whereby intake was restricted in TN sheep to match that of HE animals in an attempt to remove any confounding effects of dissimilar nutrient intake [7,30,31]. Sheep exposed to heat (HE) were given *ad libitum* access to feed, and their refusals were measured once daily by weighing orts. The amount (weight) of feed consumed by the HE pair member was then offered to their randomly selected TN partner on the following day, and TN intake was also measured by weighing orts. Water was offered *ad libitum*, and water intake was measured twice daily (at feeding times) by measuring the volume consumed (in L) from the water bucket.

Sheep were housed in metabolism crates within climate-controlled rooms. The TN environment was maintained at 21 (±0.7) °C and 57 (±7.7)% relative humidity (RH). The HE environment protocol began at 0900 h, with the temperature gradually increasing (15 °C per h) to reach a maximum of 43 °C and a minimum of 49 (±11.8)% RH, where it remained until 1700 h, at which point the temperature was returned to and maintained at TN conditions overnight. Due to fluctuations in the room temperature, the average temperature throughout the HE protocol was 37 (±3.5) °C.

### 2.2. Metabolic Challenges

After 21 days of treatment, sheep were subjected to glucose (intravenous glucose tolerance test, IVGTT; day 21), insulin (ITT; day 22), and ACTH (day 23) challenges, commencing at 1000 h daily. Chronic indwelling catheters were fitted into the external jugular 2 h prior to the IVGTT (at 0800 h) on day 21 to reduce the stress associated with repeated blood collections. Briefly, the sheep was restrained, and a small area of the neck (over the jugular vein) was clipped free of wool and cleaned with 70% ethanol. The vein was then located visually, which confirmed the occlusion of the jugular furrow. A 14-gauge, 3.25-inch angiocath catheter (BD, Macquarie Park, NSW, Australia) was inserted into the vein approximately 8 cm deep and then secured to the skin using medical tape and superglue. A 22-centimeter plastic catheter extension with a luer lock (Heidelberg extension tubing, B.Braun, Breinigsville, PA, USA) pre-filled with heparinized saline (50 U/L) was secured to the catheter. The catheter was flushed with 8–10 mL of heparinized saline (50 U/L) and sealed with a Safesite (B.Braun). The insertion site and catheter tubing were covered in a 10-centimeter cohesive bandage, and the catheter tubing was tucked into the bandage. When not frequently in use, the patency of the catheter was maintained by flushing with 8–10 mL of heparinized saline (50 U/L) twice daily. Blood samples were collected via the jugular catheter into a 10 mL lithium heparin vacutainer (BD). Blood samples were stored on ice immediately after collection. Plasma was harvested via centrifugation at 1000× *g* for 10 min at 4 °C. The obtained plasma was stored at −20 °C prior to sample analysis. During blood sampling procedures, the catheter was flushed with heparinized saline (25 U/L) immediately after every blood sample collection. Catheters were removed following the ACTH challenge on day 23.

Feed was removed from sheep 12 h prior to each challenge, and sheep remained under temperature treatment (TN or HS) during the challenge periods. Sheep were fed their allocated feed and a daily dose of dietary betaine with their feed once the challenge was completed. For the IVGTT, glucose (0.3 g/kg glucose on a liveweight basis, given as a 40% glucose (dextrose) solution) was administered via the jugular catheter and immediately flushed with 20 mL sterile saline. Blood samples were collected via the catheter at −30, −15, and −1 min (relative to glucose infusion) to establish baseline levels for each challenge, then at 2, 3, 4, 5, 6, 8, 10, 12, 14, 16, 18, 20, 22, 25, 30, 35, 40, 45, 50, 60, 75, 90, 120, 150, 180, 210, and 240 min for the IVGTT. For the ITT, insulin (ActRapid Human Insulin 100 IU/mL; Novo Nordisk Pharmaceuticals Pty Ltd., Sydney, NSW, Australia) was infused intravenously via the jugular catheter at a dose of 0.125 U/kg liveweight, followed by 10 mL sterile saline. Blood samples were obtained at −30, −15, −1, 5, 10, 15, 20, 25, 30, 35, 40, 45, 50, 55, 60, 65, 70, 75, 80, 85, 90, 95, 100, 105, 110, 115, 120, 150, 180, 210, 240, 270, and 300 min relative to the infusion. The ACTH was given at a dose of 2 µg/kg liveweight (mixed to 1 mL with sterile saline, Synacthen, Novartis, Macquarie Park, NSW, Australia) as chosen based on previous work [32,33] and injected intramuscularly into the rump of the animal with a 20 g needle. Blood samples were obtained via the catheter at −30, −15, −1, 30, and 60 min relative to the ACTH infusion.

Plasma samples were analyzed for glucose, NEFA, insulin (IVGTT samples), and cortisol (ACTH samples) concentrations. Plasma NEFA concentration was measured using a kit assay (Wako C NEFA kit, Novachem, Heidelberg West, VIC, Australia) as modified by Johnson and Peters [34]. Plasma glucose concentration was measured using the Infinity glucose oxidase liquid stable reagent (Thermo Scientific, Noble Park, VIC, Australia) as per the manufacturer’s instructions. Inter and intra-assay coefficients of variation were 5.3 and 8.7% for glucose and 4.7 and 7.4% for NEFA. Plasma insulin concentration was measured using the double antibody radioimmunoassay (RIA) technique (Millipore porcine insulin RIA, Abacus ALS, QLD) as described by [35,36,37]. Inter and intra-assay coefficients of variation for low and high QCs within the insulin assay were 2.5 and 2.0%, and 4.7 and 34.6%, respectively. Plasma cortisol was measured in duplicate via radioimmunoassay (RIA) with 1,2,6,7-3[H]-cortisol (TRK407, Amersham-Pharmacia Biotech, Buckinghamshire, UK), and anticortisol antiserum was donated by RI Cox and MSF Wong (CSIRO Division of Animal Production, Blacktown, NSW, Australia). Cortisol inter- and intra-assay coefficients of variation were 0.2 and 1.6%, respectively.

Plasma hormone and metabolite responses were analyzed for areas under the curve (AUC) using a linear trapezoidal summation between successive pairs of metabolite concentrations after correcting for baseline concentrations. Clearance rates (CR) are calculated as the apparent factional rate of change for a given period and are determined from the slope of the natural logarithm of the plasma concentration (of the specific hormone or metabolite) plotted against time. Timeframes for AUC calculations were measured over the same period for each sheep and are reported in the text, chosen either to capture the maximal response to the individual challenge or to report the responses for the duration of the challenge. Baseline concentrations were calculated as the mean of the samples obtained prior to infusion. Maximum (peak) and nadir values were calculated for each sheep, and mean values were reported for each specific treatment group. The mean plasma concentrations presented are the overall mean, irrespective of sample time. The revised quantitative insulin sensitivity check index (RQUICKI) was calculated from the responses to the IVGTT using the formula RQUICKI = 1/log fasting insulin (µU/mL) + log fasting glucose (mg/dL) + log fasting NEFA (mmol/L) [38].

Results from the IVGTT were subjected to Bergman’s minimal modeling [39,40], using MINMOD Millennium (2001) [41]. The MINMOD Millennium (referred to as MINMOD from here on) computer model estimation has been validated for use in humans, rats, pigs, dogs, and cats [42], and although data suggests that this model may be a useful tool in the assessment of insulin sensitivity in cows [37,43,44], dairy goats [45], and sheep [46], it has not been validated for use in these species. This model is used to describe the nature of the interactions between glucose and insulin actions and kinetics. This model is also used to calculate the effect of insulin on accelerating glucose disposal within the body following a glucose challenge, or “insulin sensitivity”. The MINMOD model uncouples the glucose and insulin responses and expands them into two sub-systems: the glucose minimal model, which uses insulin observations to drive the glucose response, and the insulin minimal model, which uses glucose observations to drive the insulin response. These responses are then used to portray glucose dynamics, such as insulin sensitivity and glucose responsiveness. The output parameters and indices derived from the MINMOD Millennium software are: SI = insulin sensitivity ((mU/L)^−1^ min^−1^), which quantifies the capacity of insulin to promote glucose disposal through the GLUT4 receptors and to inhibit the endogenous production of glucose; Sg = glucose effectiveness (min^−1^), capacity of glucose to mediate its own disposal through GLUT1 receptors; AIRg = acute insulin response to glucose (mU L^−1^ min^−1^), addresses the adequacy of insulin secretion through pancreatic β-cell function (defined as insulin AUC between 0–10 min) and as such, is a measure of pancreatic responsivity; DI = disposition index (AIRg × SI), which combines the information on the individual contributions of insulin sensitivity and pancreatic responsivity to give the speed at which the subject responds to the glucose challenge; Ib = (mU L^−1^), basal insulin concentration pre-challenge; Gb = (mg/dL^−1^), basal glucose concentration pre challenge; and insulin resistance = (mM mU L^−2^), where decreased values for insulin resistance correspond to increased insulin sensitivity and vice versa, calculated by the equation: basal glucose (mM) × basal insulin (µU)/22.5.

### 2.3. Tissue Biopsies

On day 24, sheep were biopsied, and samples of skeletal muscle (*longissimus dorsi*) and subcutaneous adipose tissue (0.5–1 g of each tissue) were obtained between the 2nd and 4th lumbar vertebrae, midway between the spine and the end of the transverse process. Animals were placed under general anesthesia using 0.1 mg/kg liveweight intramuscular xylazine (Troy Laboratories, Glendenning, NSW, Australia) as a sedative and muscle relaxant given 10 min prior to anesthesia, followed by 5 mg/kg intravenous ketamine (Troy Laboratories). Biopsy procedures were undertaken under aseptic conditions and completed within 15 min. The internal biopsy site was closed using an absorbable 2-0 catgut suture, and the external site was closed with 1-0 non-absorbable silk black sutures and covered with topical Trycin antibiotic powder (Juroz Pty Ltd., Wedderburn, NSW, Australia). External sutures were removed 14 days post-biopsy. Samples were snap-frozen immediately in liquid nitrogen and then stored at −80 °C prior to analysis.

### 2.4. mRNA Extraction and RT-PCR Analysis

Tissue samples were subjected to RT-PCR for mRNA gene expression analysis of β-actin (housekeeper), HSP70, HSP90, and adenosine monophosphate kinase (AMPK) in skeletal muscle and HSP70/90, peroxisome proliferator-activated receptor γ (PPARγ), hormone-sensitive lipase (HSL), leptin, adiponectin, and fatty acid synthase (FAS) in adipose tissue. Primers (Table 1) were designed using the OlgioPerfect Designer program (Invitrogen, Waltham, MA, USA) and obtained from GeneWorks Pty Ltd. (Hindmarsh, SA, Australia). Primers were diluted to the appropriate concentration in Tris-EDTA (TE) buffer (10 mM Tris-HCl, 1 mM EDTA, pH 8.0), depending on the concentration of the primer ordered. Frozen tissue samples were ground into a fine powder using liquid nitrogen and a mortar and pestle, and then mRNA was extracted using Trizol (Life Technologies Australia Pty Ltd., Mulgrave, VIC, Australia) and a PureLink RNA Mini Kit (Ambion, Life Technologies Australia Pty Ltd., Tullamarine, VIC, Australia) as per the manufacturer’s protocol. The quality and quantity of RNA extracted from the tissue samples were determined using the Experion System automated electrophoresis station (Bio-Rad Laboratories Inc., Gladesville, NSW, Australia) and the Experion StdSens Analysis Kit (Bio-Rad Laboratories Inc., Gladesville, NSW, Australia). The electropherograms for each sample were analyzed for the concentration of RNA and the ratio of 28S to 18S. Samples with two clear peaks and a ratio close to 1.7 were accepted (mean ratio was 8.6), and all others were discarded and re-extracted. Following quality and quantity evaluation of the RNA, total RNA (8 μL) was transcribed to cDNA using the SuperScript III First-Strand Synthesis System for RT-PCR (Invitrogen) as per the manufacturer’s instructions using a Corbett Palm Cycler PCR (Corbett Research, Mortlake, VIC, Australia). One µL from each cDNA reaction was added to a common pool to be used as a positive control for each RT-PCR reaction.

The RT-PCR reactions were undertaken in triplicate wells of 23 μL (12.5 µL SYBR green (Bio-Rad Laboratories, Gladesville, NSW, Australia), 8.5 µL DEPC water, 1 µL each of forward and reverse primers, and 2 μL cDNA) in a 96-well PCR plate, the plate covered in film, and then briefly centrifuged. The RT-PCR was then completed using the iQ5 system (BioRad). Each plate contained a non-template control and a positive control (pooled cDNA). A standard curve (5 serial dilutions of a pooled cDNA sample) was run for each gene to determine the amplification efficiency of each. The following general cycling protocol was used for each PCR reaction: Cycle 1: initial denaturation 95 °C for 3 min; Cycle 2: (repeated 40 times) 95 °C for 10–30 s, 60–61 °C for 30 s (temperature dependent on primer), then 72 °C for 30 s (optional step dependent on primer); Cycle 3: 95 °C for 1 min; Cycle 4: 60 °C for 1 min; Cycle 5: (repeated 71 times) 60 °C for 30 s. A melt curve was produced for each run to determine if any primer-dimers were produced. If primer dimers were present, the run was discarded and re-run using fresh primer stock. Analysis was performed using the iQ5 Optical System Software version 2.0 (Bio-Rad Laboratories). The RT-PCR data was analyzed as the threshold cycle (Ct) relative to that of the housekeeping gene β-actin. A difference in ∆Ct of −1.0 is associated with a doubling (200%) and a halving (50%) of gene expression. For ease of presentation, the data are also presented in parenthesis as % of a reference sample (TN, 0 g betaine/day).

### 2.5. Statistical Analysis

Plasma hormone and metabolite responses were subjected to the restricted maximum likelihood (REML) analysis function suitable for repeated measures in GenStat (19th edition, VSN International Ltd., Hemel Hampstead, UK). The main effects and all interactions between temperature, dietary treatment, and time (relative to infusion) were included, with the animal included as the random factor and blocked for the effects of replication. In addition, the overall effects of betaine were assessed (pooled for 2 + 4 g betaine/day) and reported, as were the dose responses (2 vs. 4 g betaine/day). Basal values were pooled across each challenge day and presented as pooled values where relevant. Homogeneity of the resultant hormone and metabolite data was tested by generating residual plots and heterogeneous data, and as a result, all responses to metabolic challenges were log transformed prior to statistical analysis. Reported values are back-transformed and presented in all figures. All results presented are predicted means ± standard error of differences (SED) unless otherwise stated. Data reported as basal or time ‘0’ is the mean of each of the baseline samples collected. Significance was declared at *p* < 0.05. Non-significant responses or interactions are not reported. Physiological responses to HE have been previously presented by DiGiacomo et al. [27] and will not be reported in full here.

## 3. Results

### 3.1. Metabolic Challenges

Responses to the three metabolic challenges are presented in Table 2, Table 3, Table 4, Table 5 and Table 6 and Figure 1, Figure 2 and Figure 3 as back-transformed data. As there were few instances where the betaine dose significantly impacted the reported measures, the data is presented as pooled betaine (2 + 4 g/day) unless otherwise stated. Non-significant measurements (e.g., CR and AUC calculations) and interactions are not reported.

Pooled (across challenge days) basal responses are presented in Table 2. There was no effect of temperature on basal plasma glucose (*p* = 0.73), insulin (*p* = 0.069), or NEFA (*p* = 0.56) concentrations, while the ratio of insulin to glucose was greater in HE animals (1.67 vs. 2.98 mU/mmol for TN and HE, respectively, SED 0.5827, *p* = 0.069). The HOMA-IR was greater (1.15 vs. 1.91, SED 0.333, *p* = 0.029), and the RQUICKI (0.340 vs. 0.361, SED 0.0158, *p* = 0.021) was reduced in HE sheep. There was no effect of betaine on basal plasma glucose (*p* = 0.90) or cortisol concentrations. Basal insulin was greater (4.52 vs. 8.13 mU/L for control vs. betaine, SED 1.256, *p* = 0.017) in sheep supplemented with betaine (pooled 2 + 4 g/day), while there was no difference between betaine doses (2 vs. 4 g/day, *p* = 0.79). There was no effect of betaine on the ratio insulin:glucose (*p* = 0.11). Basal NEFA was lower in sheep fed betaine (255 vs. 191 mmol/L, SED 1.145, *p* = 0.05), while there was no difference between betaine doses (*p* = 0.065). Betaine decreased the RQUICKI (0.264 vs. 0.236, SED 0.0079, *p* = 0.002) compared to control. There were no significant interactions between temperature and betaine for any of the basal measures (Table 2).
animals-13-01691-t002_Table 2Table 2Basal plasma glucose, insulin, and NEFA concentrations and estimated homeostatic model assessment of insulin resistance (HOMA-IR) and revised quantitative insulin sensitivity index (RQUICKI) responses in female Merino sheep (*n* = 36) fed either 0, 2, or 4 g betaine/day and exposed to either thermoneutral (TN) or heat conditions ^1^.
ThermoneutralHeat Exposure
*p*-ValueBetaine (g/day)024024SED ^2^Temp ^3^Betaine ^4^
Within Betaine ^5^Glucose, mmol/L3.653.653.903.873.733.831.0960.730.900.56Insulin, mU/L3.168.375.896.448.2710.61.4480.0690.0170.79Insulin:glucose, mU/mmol1.042.351.632.293.283.371.0090.0330.110.69NEFA, mmol/L 2832171692292071751.2470.560.0360.065HOMA-IR0.5751.731.141.581.982.170.57590.0290.0610.63RQUICKI0.4580.3610.3800.3820.3570.3440.02740.0150.0010.92^1^ Data are pooled across the glucose (IVGTT), insulin (ITT), and ACTH challenges. There were no interactions between temperature and betaine, so the main effects are presented. ^2^ Standard error of the difference for the effect of temperature × betaine. ^3^ Thermoneutral vs. heat exposure (*n* = 18 vs. 18). ^4^ Control vs. betaine (*n* = 12 vs. 24). ^5^ 2 vs. 4 g/head/day (*n* = 12 vs. 12).


#### 3.1.1. IVGTT

The responses to the IVGTT are reported in Table 3. Temperature did not impact basal (IVGTT day) plasma glucose (*p* = 0.51) or NEFA (*p* = 0.56) concentrations, while the basal plasma insulin concentration was greater in HE-treated sheep (4.48 vs. 8.17 mU/L, SED 1.265, *p* = 0.026). As demonstrated in Figure 1, there were significant interactions between temperature × time (*p* = 0.039) and betaine × time (*p* = 0.004). There were no main effects of temperature on any of the IVGTT responses (Table 3), except for the insulin AUC_0–240min_, which was reduced in HE sheep (5207 vs. 2491 mU.min/L, SED 1207, *p* = 0.048). Pooled betaine increased the basal insulin concentration (4.29 vs. 8.54, SED 1.26, *p* = 0.006). The insulin AUC_0–5min_ was greater in sheep fed betaine (118 vs. 178 mU.min/L, SED 26.61, *p* = 0.032), and there was an interaction such that the insulin AUC_0–5min_ was lowest in TN control sheep and greater in all other treatments (Table 3). The ratio of basal glucose to insulin tended to be reduced by HE (0.82 vs. 0.48, *p* = 0.055) and was greater for control sheep (0.85 vs. 0.47, SED 1.285, *p* = 0.024), though there was no interaction between temperature and diet (*p* = 0.45).

Plasma concentrations of insulin increased rapidly in response to the glucose infusion, while plasma NEFA concentrations began decreasing approx. 5 min after the glucose infusion. The peak insulin concentration observed (86 mU/L, *p* = 0.21 for the interaction) and the time taken to reach this peak (approx. 17 min, *p* = 0.20 for the interaction) were not different between treatments. The insulin CR_6–60min_ was greater in TN sheep fed 0 g betaine/day compared to all other treatments (*p* = 0.047), Table 3. The insulin AUC_0–5min_ was greater in betaine-supplemented sheep (118 vs. 178 mU/L.min, SED 26.61, *p* = 0.032) and was greatest in TN sheep fed betaine (*p* = 0.022, Table 3). The insulin AUC_0–240min_ was greater in TN sheep compared to HE sheep (5207 vs. 2491 mU/L.min, SED 1207.0, *p* = 0.048).

There was no effect of temperature or dietary treatment on the peak plasma glucose (*p* = 0.78 and 0.21, respectively), NEFA (*p* = 0.64 and 0.81), or insulin (*p* = 0.79 and 0.13) concentrations in response to the IVGTT. The glucose AUC did not differ due to temperature (*p* > 0.58) or diet (*p* > 0.33) for any period throughout the IVGTT.
Figure 1Plasma glucose, insulin, and NEFA responses to an intravenous glucose tolerance test (IVGTT; 0.3 g/kg liveweight) in sheep (Merino ewes, *n* = 6 sheep per group) exposed to thermoneutral (TN) conditions (blue solid lines) fed 0 (○) or combined (2 + 4 g betaine/day; □) and 0 (●), or combined (2 + 4 g betaine/day; ■) in sheep exposed to heat (HE) conditions (red dashed lines). The inserts are the period from 0–30 min. The standard errors for the difference in the interaction between temperature and dietary betaine treatment (0 vs. 2 + 4 g/day; pooled across the whole challenge period) are (**A**) 1.137 mM; (**B**) 1.380 mU; and (**C**) 1.265 μM. The *p*-values for the effects of temperature, time, betaine (2 + 4 g/day), temp. ×. betaine, temp. ×. time, time. ×. Betaine, and temp. ×. betaine × time are (**A**) 0.217, <0.001, 0.637, 0.950, 0.988, 0.253, and 0.986; (**B**) 0.948, <0.001, 0.114, 0.238, 0.039, 0.004, and 0.069; (**C**) 0.118, <0.001, 0.005, 0.252, 0.935, 0.861, and 0.996.
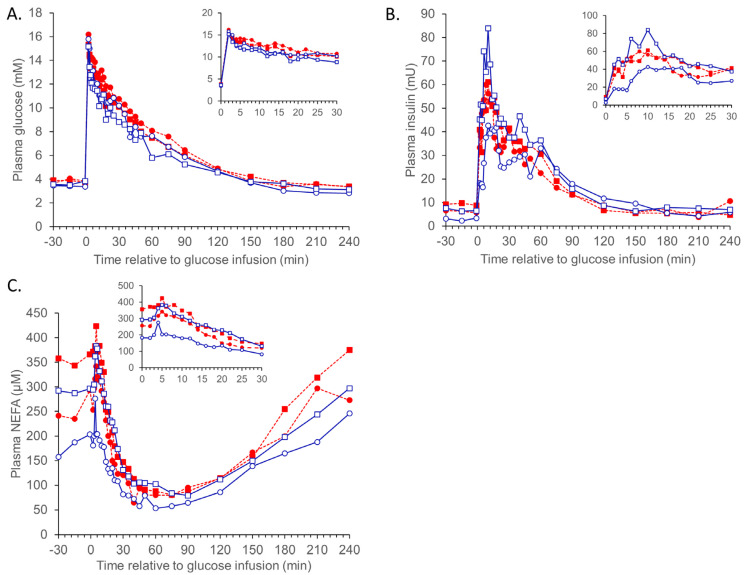

animals-13-01691-t003_Table 3Table 3Plasma glucose, insulin, and NEFA responses to a glucose (IVGTT) challenge in female Merino sheep (*n* = 36) fed either 0, 2, or 4 g betaine/day and exposed to either thermoneutral (TN) or heat conditions ^1^.
ThermoneutralHeat Exposure
*p*-ValueBetaine (g/day)024024SED ^2^Temp ^3^Betaine ^4^Within Betaine ^5^Glucose baseline (mM)3.433.393.794.113.334.101.3000.510.210.85Glucose peak (mM)15.917.616.416.417.117.21.1060.780.330.67Glucose CR_6–30_ % min^−1^
−0.004−0.004−0.007−0.005−0.005−0.0070.00120.190.210.009Insulin baseline (mU/L)3.007.446.137.699.9112.21.5500.0260.0060.73G:I_Baseline_
1.220.4670.6630.5910.4240.3661.5180.0550.0240.85Insulin peak (mU/L)59.812480.585.383.896.21.5310.790.130.42Insulin AUC_0–5_ (mU/L.min)71.124714416414717340.250.780.0320.14Insulin AUC_0–20_ (mU/L.min)664137988210677661251285.80.470.300.35Insulin AUC_0–240_ (mU/L.min)6451379230342635236223331931.00.0480.260.23Insulin CR_6–60min_ % min^−1^
−0.002−0.006−0.004−0.007−0.002−0.0080.00220.460.450.64NEFA baseline (µM)3512602463423602481.3490.560.220.16NEFA peak (µM)5375413134935413132.5840.640.810.99NEFA nadir (µM)45.161.853.263.157.543.11.2730.970.13<0.001CR = clearance rate; AUC = area under the curve (corrected for basal levels). ^1^ There were generally no interactions between temperature and betaine, so the main effects are presented. Temperature × Betaine *p*-Values (where significant): Insulin AUC_0–5_ (mU/L.min) *p* = 0.022; Insulin CR_6–60min_ % min^−1^ *p* = 0.047. ^2^ Standard error of the difference for the effect of temperature × betaine. ^3^ Thermoneutral vs. heat exposure (*n* = 18 vs. 18). ^4^ Control vs. betaine (*n* = 12 vs. 24). ^5^ 2 vs. 4 g/head/day (*n* = 12 vs. 12).


The indices calculated from minimal modeling are presented in Table 4. While basal glucose concentrations calculated from the model were unaffected by temperature (*p* = 51) or betaine (*p* = 0.20), the Ib concentration was greater in sheep fed dietary betaine (4.86 vs. 7.59 mU/L, SED 1.253, *p* = 0.037). The calculated insulin resistance was greater in sheep fed betaine (0.77 vs. 1.28 (mM·mU/L)2, SED 0.199, *p* = 0.015). The SI was lower in sheep fed betaine (8.48 vs. 4.77 (mU/L)-1·min-1, SED 1.756, *p* = 0.041). There was a significant interaction between temperature and diet such that the DI was lowest in TN control sheep and greatest in HS control diet sheep (Table 4, *p* = 0.053). No further indices calculated by the model were altered by temperature or dietary betaine (Table 4).
animals-13-01691-t004_Table 4Table 4Minimal modeling (MINMOD) responses to an intravenous glucose tolerance test in female Merino sheep (*n* = 36) fed either 0, 2, or 4 g betaine/day and exposed to either thermoneutral or heat conditions ^1^.
Thermoneutral Heat Exposure 
*p*-Value Betaine (g/day)024024SED ^2^Temp ^3^Betaine ^4^Within Betaine ^5^Gb (mg/dL)58.770.968.565.972.772.89.7560.510.200.60Ib (mU/L)3.337.406.086.397.299.701.7660.090.0370.99Insulin resistance ((mM.mU/L)^2^)0.481.291.041.051.151.650.28030.090.0150.48AIRg ((mU/L)·min^−1^)16236424227120530872.670.970.230.44SI ((mU/L)^−1^·min^−1^)6.525.204.0310.434.914.732.4780.430.0410.33DI1070200297126789851354697.00.490.260.33Sg0.0100.0090.0180.0110.0130.0130.00400.880.330.28Gb: basal glucose concentration; Ib: basal insulin concentration; AIRg: acute insulin response to glucose; SI: insulin sensitivity; DI: disposition index; Sg: glucose effectiveness. ^1^ There were generally no interactions between temperature and betaine, so the main effects are presented. Temperature × Betaine *p*-Values (where significant): DI *p* = 0.053. ^2^ Standard error of the difference for the effect of temperature × betaine. ^3^ Thermoneutral vs. heat exposure (*n* = 18 vs. 18). ^4^ Control vs. betaine (*n* = 12 vs. 24). ^5^ 2 vs. 4 g/head/day (*n* = 12 vs. 12).


#### 3.1.2. ITT

The responses to the ITT are reported in Table 5. There was no effect of temperature or diet on basal plasma glucose or NEFA basal, peak, or nadir responses to the ITT (Table 5). The effects of temperature or dietary treatment on glucose and NEFA AUC and CR responses to insulin (Figure 2) were not significant and are thus not presented in the table.
Figure 2Plasma glucose (**A**) and NEFA (**B**) responses to an insulin tolerance test (ITT; 0.125 U/kg liveweight) in sheep (Merino ewes, *n* = 6 sheep per group) exposed to thermoneutral (TN) conditions (blue solid lines) fed 0 (○) or combined (2 + 4 g betaine/day; □) and 0 (●), or combined (2 + 4 g betaine/day; ■) in sheep exposed to heat (HE) conditions (red dashed lines). The inserts are the period from 0–30 min. The standard errors for the difference in the interaction between temperature and dietary betaine treatment (0 vs. 2 + 4 g/day; pooled across the whole challenge period) are (**A**) 1.114 mM and (**B**) 1.319 μM. The *p*-values for the effects of temperature, time, betaine (2 + 4 g/day), temp. ×. betaine, temp. ×. time, time. ×. Betaine, and temp. ×. betaine × time are (**A**) 0.828, <0.001, 0.716, 0.441, 0.455, 0.808, and 0.469; and (**B**) 0.324, <0.001, 0.512, 0.101, 0.146, 0.875, and 0.227.
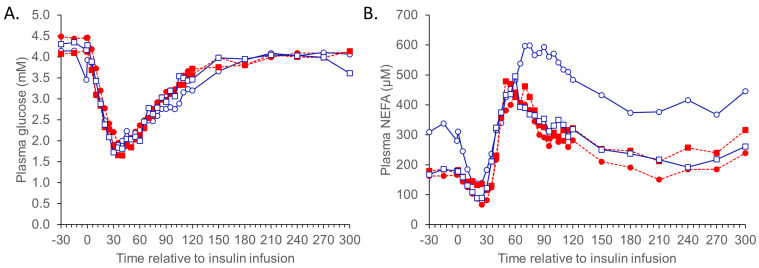

animals-13-01691-t005_Table 5Table 5Plasma glucose, insulin, and NEFA responses to an insulin tolerance test (ITT) in female Merino sheep (*n* = 36) fed either 0, 2, or 4 g betaine/day and exposed to either thermoneutral (TN) or heat conditions ^1^.
ThermoneutralHeat Exposure
*p*-Value024024SED ^2^Temp ^3^Betaine ^4^Within Betaine ^5^Glucose baseline (mM)3.934.204.364.464.323.901.1020.800.990.62Glucose nadir (mM)1.611.361.551.731.621.231.1890.980.150.55Glucose peak (mM)4.504.284.694.464.544.141.1080.660.810.67NEFA baseline (µM)2342161471681451471.3760.220.310.37NEFA peak (µM)7806525135026175221.2880.280.570.23NEFA nadir (µM)10464.763.550.986.177.11.5270.740.970.82^1^ There were no interactions between temperature and betaine, so the main effects are presented. ^2^ Standard error of the difference for the effect of temperature × betaine. ^3^ Thermoneutral vs. heat exposure (*n* = 18 vs. 18). ^4^ Control vs. betaine (*n* = 12 vs. 24). ^5^ 2 vs. 4 g/head/day (*n* = 12 vs. 12).


#### 3.1.3. ACTH

The responses to the IVGTT are reported in Table 6. Basal plasma cortisol (*p* = 0.29 and 0.23) and glucose (*p* = 0.77 and 0.80) concentrations did not differ due to temperature or diet treatments. While there was no influence of temperature (*p* = 0.52), basal plasma NEFA was reduced in sheep supplemented with betaine (239 vs. 138 µM, SED 1.24, *p* = 0.019).

There was no difference in plasma cortisol concentrations (or associated AUC and CR values) in response to the ACTH infusion due to either temperature or dietary betaine (Figure 3). The peak plasma NEFA was reduced in sheep fed betaine (303 vs. 173 µM, SED 1.21, *p* = 0.007). No other significant responses to the ACTH challenge were observed.
animals-13-01691-t006_Table 6Table 6Plasma cortisol, NEFA, and glucose responses to an ACTH challenge in female Merino sheep (*n* = 36) fed either 0, 2, or 4 g betaine/day and exposed to either thermoneutral (TN) or heat conditions ^1^.
ThermoneutralHeat Exposure
*p*-Value024024SED ^2^Temp ^3^Betaine ^4^Within Betaine ^5^Cortisol baseline (µM)6.682.954.355.504.966.541.4840.290.230.22Cortisol peak (µM)11396.210612812695.11.2630.480.340.55NEFA baseline (µM)2831661192021341371.4440.520.0190.56NEFA peak (µM)3482271622631571571.3750.280.0070.50Glucose baseline (mM)3.593.203.243.123.423.251.2100.770.800.88Glucose peak (mM)3.813.793.583.273.583.901.1380.580.520.89^1^ There were no interactions between temperature and betaine, so the main effects are presented. ^2^ Standard error of the difference for the effect of temperature × betaine. ^3^ Thermoneutral vs. heat exposure (*n* = 18 vs. 18). ^4^ Control vs. betaine (*n* = 12 vs. 24). ^5^ 2 vs. 4 g/head/day (*n* = 12 vs. 12).
Figure 3Plasma cortisol (**A**), glucose (**B**), and NEFA (**C**) responses to an ACTH (2 µg/kg liveweight) infusion in sheep (Merino ewes, *n* = 6 sheep per group) exposed to thermoneutral (TN) conditions (blue solid lines) fed 0 (○) or combined (2 + 4 g betaine/day; □) and 0 (●), or combined (2 + 4 g betaine/day; ■) in sheep exposed to heat (HE) conditions (red dashed lines). The standard errors for the difference in the interaction between temperature and dietary betaine treatment (0 vs. 2 + 4 g/day; pooled across the whole challenge period) are (**A**) 1.339 ng/mL; (**B**) 1.119 mM; and (**C**) 1.316 μM. The *p*-values for the effects of temperature, time, betaine (2 + 4 g/day), temp. ×. Betaine, temp. ×. Time, time. ×. betaine, and temp. ×. Betaine × time are (**A**) 0.248, <0.001, 0.299, 0.680, 0.534, 0.700, and 0.895; (**B**) 0.657, 0.009, 0.755, 0.142, 0.858, 0.115, and 0.456; and (**C**) 0.417, 0.882, 0.016, 0.616, 0.568, 0.742, and 0.582.
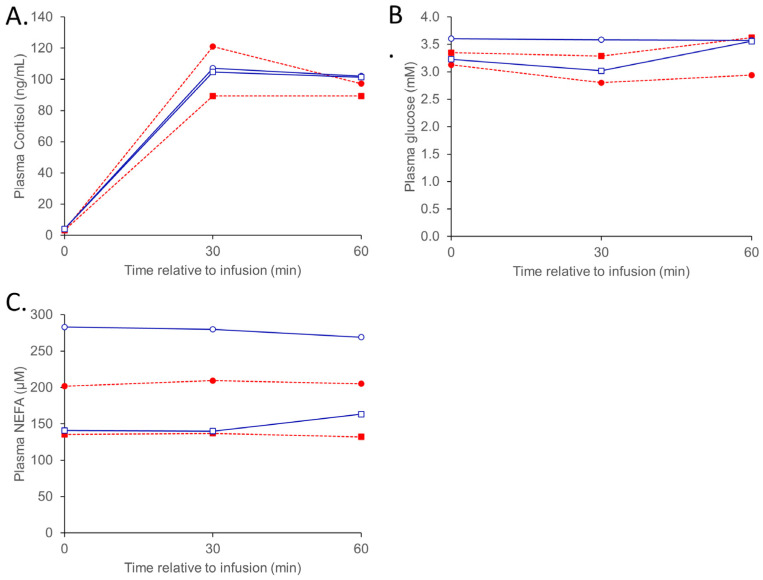



### 3.2. Tissue Gene Expression

Adipose and muscle tissue gene expression results are presented in Table 7. Tissue type did not alter the expression of *HSP70* (*p* = 0.32), while there was increased expression of *HSP90* in adipose tissue compared with skeletal muscle tissue (*p* < 0.001, Table 7). None of the expressions of genes measured in either adipose or muscle tissue were altered by heat, dietary betaine supplementation, or their interaction (Table 7). The expression of *HSP90* (*p* = 0.36) and *FAS* (*p* = 0.29) in adipose tissue did not differ due to betaine treatment overall (2 + 4 g/day), but there was a within-betaine dose response such that *HSP90* (−3.16 vs. −3.84 ∆Ct, +60%, *p* = 0.03) expression was greater and *FAS* (0.33 vs. 1.77 ∆Ct, −63%, *p* = 0.01) lesser in sheep fed 4 compared to 2 g/day betaine.

## 4. Discussion

The major finding from this study was that dietary betaine decreased lipid mobilization, as evidenced by plasma NEFA. Furthermore, HE tended to decrease plasma NEFA, particularly in sheep not consuming betaine. Basal insulin was increased by both HE and dietary betaine, and given that insulin is antilipolytic, this may provide an insight into its mechanism of action. In the same experiment, we previously demonstrated that betaine supplementation (both 2 and 4 g/day) decreased circulating plasma NEFA (measured weekly) by ~25 μM [27]. In the present experiment, sheep fed betaine had a reduced basal NEFA concentration (mean across all challenge days) and a reduced peak NEFA response to the ACTH infusion. This supports our previous findings and suggests that betaine is impacting lipid metabolism, although this response is not related to differences in sensitivity of the hypothalamo-pituitary-adrenal axis, as plasma cortisol responses to ACTH did not differ in our experiment. Finally, as plasma glucose responses were not altered by betaine supplementation (both in the challenges and in the basal measures), the changes in plasma NEFA were not occurring in direct response to circulating glucose and may be in response to increased insulin inhibiting NEFA production, as supported by a greater insulin baseline measured in betaine-supplemented sheep during the IVGTT. This further suggests that dietary betaine responses occur metabolically rather than via changes to direct stress responses. Studies have shown that serum glucose was increased in betaine-supplemented and heat-stressed dairy cattle [57 and 114 mg/kg BW [47]] and [15 and 30 g/cow/day [48]], while serum NEFA and β-hydroxybutyrate (BHB) were decreased in betaine-supplemented cows, as betaine improved nutrient intake and digestibility, leading to an increase in available energy [48].

In the present study, we showed that the RQUICKI values were reduced and the calculated HOMA-IR tended to be greater in sheep fed betaine, indicating decreased insulin sensitivity in these animals. This is further supported by the MINMOD responses, whereby SI, Ib, and IR were all increased in sheep fed betaine. To the best of our knowledge, there are no published studies directly investigating the metabolic responses to betaine supplementation in ruminants, but our data supports the findings in other species and models that betaine impacts adipose tissue function. Wang et al. [49] showed an improved response to an IVGTT in mice supplemented with betaine and fed a high-fat diet (betaine reduced HOMA-IR, prevented severe hyperglycemia in response to glucose, and decreased circulating NEFA). These authors also found that betaine supplementation attenuated changes in circulating adiponectin, resistin, and leptin concentrations triggered by the high-fat diet while also improving insulin sensitivity and reducing endoplasmic reticulum stress at the adipose tissue level [49]. As reviewed by Arumugam et al. [50], betaine supplementation can attenuate or prevent liver injury, preserve adipose tissue function, reduce oxidative and endoplasmic reticulum stress, and reduce inflammation. The gene expression of *FAS*, which catalyzes the final step in the lipogenic (fatty acid synthesis) pathway, was reduced in the adipose tissue of sheep fed 4 compared to 2 g betaine/day. In pigs, dietary betaine supplementation reduced the mRNA expression and activity of *FAS* [51]. As FAS is a key enzyme involved in fatty acid synthesis, this would appear to be linked to alterations in adipose tissue lipid synthesis; however, due to the limited amount of tissue able to be obtained from the biopsy in the present experiment, FAS enzyme activity was not able to be measured, and therefore it cannot be concluded that altered *FAS* mRNA expression translates to enzyme activity differences. However, in this experiment, there were no differences in adipose tissue *leptin*, *adiponectin HSL*, or *PPARγ* mRNA expressions in HE or betaine-supplemented sheep.

While this experiment demonstrated reductions in circulating basal NEFA, this was not supported by the MINMOD results, which indicated that sheep supplemented with betaine had increased insulin resistance. This may indicate that betaine is improving insulin signaling and therefore there is no need to increase pancreatic insulin secretion, although this is not supported by the present experiment whereby betaine increased basal insulin concentrations. Increased circulating NEFA concentrations are involved in insulin resistance (by negatively impacting the insulin secretory capacity of the pancreas), and therefore a decrease in circulating NEFA due to dietary betaine supplementation may be beneficial to heat-stressed ruminants. Dietary betaine supplementation reverses insulin resistance in an in vitro model, and in both in vitro and in vivo models, betaine normalizes downstream signaling pathways involved in gluconeogenesis, perhaps by improving phosphorylation of early steps in the insulin signaling cascade [52]. As the sheep fed betaine had a greater initial insulin AUC_0–5min_ and the glucose to insulin baseline ratio was greater for control sheep, coupled with the increase in insulin resistance for betaine supplemented sheep calculated by MINMOD, it appears that dietary betaine increases pancreatic insulin secretion under TN conditions but less so during HE. This may be related to the impact of heat on insulin sensitivity.

In this experiment and that of Huang et al. [53], heat decreased the insulin AUC during the IVGTT (0–240 min). In small ruminants (sheep and goats), homeorhetic adaptations that suppress lipid mobilization and increase protein catabolism in response to heat stress [54,55] have been reported. Specifically, a greater peak glucose and glucose clearance rate, a lower glucose half-life, and no change in plasma insulin were observed in response to an IVGTT in heat-stressed lactating ewes [56]. They also noted no difference in responses to an ITT due to heat, while heat-stressed ewes had a lower plasma NEFA response and a more rapid NEFA clearance rate in response to an epinephrine challenge [56]. Together, these responses suggest heat-stressed lactating ewes were more resistant to lipolytic signals compared to TN ewes. It remains unknown why insulin concentrations are increased in heated ruminants, although it is thought to be part of the general processes allowing animals to acclimate to heat [57]. Compared to lactating cattle, the physiological and metabolic impacts of heat stress on sheep and other small ruminants do not appear to be as severe, which might explain the lack of significant metabolic challenge response differences between temperature treatments presented in this experiment. This is likely driven by the decreased metabolic heat production in small compared to large ruminants (particularly those producing milk) and is supported by the lack of change in plasma insulin noted in heat-stressed ewes [53,56] and dairy goats [50].

In ruminants, insulin resistance is associated with impaired insulin binding to adipocytes [54] rather than an impairment of pancreatic function. This is partially supported by the data presented here, where HE sheep tended to have greater baseline insulin concentrations and MINMOD modeling tended to show greater insulin baseline and insulin resistance, although no differences in Sg (glucose effectiveness) or SI (insulin sensitivity) were observed. This suggests that these effects are not driven by changes in glucose receptor function (GLUT4 and GLUT1). Furthermore, we did not observe any differences in glucose or fatty acids between temperature treatments in response to the ITT. We demonstrated that heat exposure reduced the RQUICKI, which is supported by the findings of Huang et al. [53], who showed that heat-stressed sheep had increased basal plasma glucose concentrations and RQUICKI, while the ratio of plasma insulin to glucose and the basal plasma NEFA concentrations were reduced. While not measured in their experiment [53] or the one presented here, our group [58] and others [18,59] have demonstrated that heat exposure increases prolactin, which may mediate the increased insulin sensitivity observed in heat-stressed ruminants.

We did not observe any differences in plasma cortisol concentrations post-ACTH infusion due to HE. Others have demonstrated an increase in plasma cortisol concentrations in response to heat stress in sheep [60], while the processes of acclimation may reduce cortisol responses to stress events. In the present experiment (as published previously), sheep reached a rectal temperature ≥40.4 °C in response to HE, and this temperature increased by 0.2 °C across the 3-week experimental period, which indicated a reduction in the animals’ ability to adapt to the chronic heat [61]. As the metabolic challenges commenced in the morning, this lack of difference in plasma cortisol may be because the animals were not yet exposed to the peak heat.

In this experiment, we also measured the expression of some key genes involved in the heat stress response in adipose and muscle tissues. Given the previously described changes to lipolysis and lipolytic signaling associated with heat stress in ruminants, it was surprising to note that there was no significant up-regulation of any of the genes measured in HE sheep. Further, although we demonstrated significant effects of dietary betaine supplementation on plasma NEFA concentrations, we did not observe any major impacts of betaine (besides the effects on *FAS* discussed previously) on tissue gene expression. Chauhan et al. [14] observed greater *HSF1* and *HSP70* but not *HSP90* mRNA expression in skeletal muscle from heat-stressed sheep. Tissue samples were only collected at one time point during this experiment, and it is thus not possible to examine temporal *HSP* gene expression changes as influenced by heat.

## 5. Conclusions

To our knowledge, this is the only experiment examining the metabolic responses to HE and dose-dependent betaine supplementation in small ruminants. Our results suggest that the positive effects elicited by betaine supplementation are likely due in part to betaine’s influence on adipose tissue metabolism and the actions of insulin. As plasma glucose responses were not altered by betaine supplementation, the noted changes in plasma fatty acids (NEFA) are likely in response to increased insulin-inhibiting fatty acid production. Betaine responses also appear to occur metabolically rather than due to direct stress responses (i.e., cortisol) and alter lipid metabolism, which supports results published in other species. This experiment provides a basis from which future experiments can be designed to further probe the metabolic responses to betaine in ruminants. Our results outline that the documented positive impacts that betaine supplementation can have on heat-stressed animals are likely due to cellular metabolic functions and energy partitioning.

## Figures and Tables

**Table 1 animals-13-01691-t001:** Details of primers used for RT-PCR in the experiment.

Gene	Abbreviation	Accession Number	Primer Sequence (5′→3′)	Annealing Temp. (°C)
Heat shock protein 72	HSP 70	U02892	F: AACATGAAGAGCGCCGTGGAGG R: GTTACACACCTGCTCCAGCTCC	61.0
Heat shock protein 90	HSP90	*	F: GACTCCCAGGCATACTGCTC R: GGCGCTGATATCTCCATGAT	60.0
β-Actin	ACTB	*	F: CATCGAGCACGGCATCGTCA R: TAGCACAGCCTGGATAGCAAC	56.0
Peroxisome proliferator-activated receptor γ	PPγ	AY 137204	F: CATAAAGTCCTTCCCGCTGA R: ACTGACACCCCTGGAAGATG	60.7
Leptin	*	AY278244	F: CGTTCTGAGGCAGTTGTTGA R: CAAATGCCTTCCCTTCAATG	58.5
Adiponectin	*	NM_174742	F: ATTATGACGGCAGCACTGG R: CCAGATGGAGGAGCACAGA	51.0
Hormone sensitive lipase	HSL	NM_001080220	F: AGCATCTTCTTTCGCACCAG R: CCGTAGAAGCAGCCTTTGTG	60.7
Fatty acid synthase	FAS	AY343889	F: CTGAGTCGGAGAACCTGGAG R: CGAAGAAGGAAGCGTCAAAC	62.2
Adenosine monophosphate kinase	AMPK	BT021145	F: CTTCCGAGCCAGTAGTCACC R: ATGCCCGTGTCCTTGTTTAG	62.2

* No abbreviation or accession number used.

**Table 7 animals-13-01691-t007:** Mean responses of adipose and skeletal muscle tissues from sheep supplemented with a 0, 2, or 4 g/day dose of dietary betaine and exposed to either thermoneutral or heat conditions ^1^. Values are ∆Ct and (% relative expression) using control thermoneutral as the standard value.

		Thermoneutral			Heat Exposure				*p*-Value		
Betaine (g/day)		0	2	4	0	2	4	SED ^2^	Temp ^3^	Betaine ^4^	Within Betaine ^5^
Adipose tissue	HSP70	−2.59	−4.04	−5.85	−4.41	−3.95	−4.16	0.949	0.68	0.44	0.22
		(100)	(273)	(954)	(352)	(256)	(296)				
	HSP90	−2.96	−3.50	−3.95	−3.50	−2.95	−3.68	0.453	0.90	0.36	0.03
		(100)	(145)	(199)	(145)	(99)	(164)				
	Leptin	−2.99	−5.57	−7.16	−5.17	−5.55	−5.65	2.062	0.86	0.19	0.58
		(100)	(600)	(1800)	(452)	(590)	(630)				
	Adiponectin	−2.85	−2.54	−1.64	−1.34	−2.90	−1.69	0.855	0.33	0.81	0.17
		(100)	(81)	(43)	(35)	(104)	(45)				
	HSL	−1.07	−1.61	−1.20	−1.55	−1.85	−1.98	0.770	0.28	0.43	0.59
		(100)	(146)	(110)	(140)	(172)	(188)				
	PPARγ	−3.83	−3.58	−3.49	−3.20	−3.98	−3.18	0.624	0.44	0.94	0.17
		(100)	(84)	(79)	(64)	(111)	(64)				
	FAS	0.39	−1.52	1.61	0.03	1.61	2.11	1.598	0.39	0.29	0.01
		(100)	(375)	(43)	(128)	(43)	(30)				
Muscle	HSP70	−4.87	−4.54	−4.85	−4.34	−4.40	−5.00	0.476	0.60	0.70	0.20
		(100)	(80)	(99)	(69)	(77)	(109)				
	HSP90	−2.33	−2.71	−2.83	−2.47	−2.34	−2.94	0.374	0.91	0.18	0.12
		(100)	(130)	(141)	(110)	(104)	(152)				
	AMPK	−8.42	−8.65	−9.01	−9.26	−8.90	−9.02	0.443	0.15	0.84	0.44
		(100)	(117)	(150)	(179)	(139)	(151)				

^1^ There were no interactions between temperature and betaine, so the main effects are presented. ^2^ Standard error of the difference for the effect of temperature × betaine. ^3^ Thermoneutral vs. heat exposure (*n* = 18 vs. 18). ^4^ Control vs. betaine (*n* = 12 vs. 24). ^5^ 2 vs. 4 g/head/day (*n* = 12 vs. 12).

## Data Availability

The data presented in this study are available on request from the corresponding author.

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
