# Peer review of "Dietary Betaine Impacts Metabolic Responses to Moderate Heat Exposure in Sheep"

_animals, 2023, doi:10.3390/ani13101691_

Round 1

Reviewer 1 Report

The study is interesting, very complete, it shows a lot of work. It has good writing. The results are interesting. However, the main problem that I notice is the Discussion of results. The authors abuse in discussing the results of other authors instead of discussing their own results. It is suggested to emphasize how betaine supplementation and heat exposure affect specifically, particularly in the variables where there was a significant effect.

It is suggested not to include the results referring to physiological response, since it would be a duplicate of the results obtained by the same authors, which is unacceptable and unethical.

It is highly recommended not to use abbreviations in the Conclusions.

There are some very backward references (eg 1, 6, 29, 35….). If possible, it would be good to replace them with more current ones.

Author Response

The study is interesting, very complete, it shows a lot of work. It has good writing. The results are interesting. However, the main problem that I notice is the Discussion of results. The authors abuse in discussing the results of other authors instead of discussing their own results. It is suggested to emphasize how betaine supplementation and heat exposure affect specifically, particularly in the variables where there was a significant effect.

The discussion has been edited and clarified to discuss the findings of this experiment more specifically.

It is suggested not to include the results referring to physiological response, since it would be a duplicate of the results obtained by the same authors, which is unacceptable and unethical.

We have removed this summary.  

It is highly recommended not to use abbreviations in the Conclusions.

Removed abbreviations.

There are some very backward references (eg 1, 6, 29, 35….). If possible, it would be good to replace them with more current ones.

We respectfully disagree that some of these older papers need to be replaced with more modern versions. Some of these seminal works set the foundations for understanding in sheep physiology and metabolism and warrant use. For example, to the best of our knowledge Mitchell et al (1979) is the only publication showing such data. However, where relevant some citations have been removed.

Reviewer 2 Report

The author has done a lot of work and carried out feeding experiments in the early stage, revealing the effect of betaine on sheep metabolism under moderate heat exposure, which is of great productive significance. But the quality of the article can be further improved. Specific suggestions for modification are as follows:

Abstract

L26-28: HOMAIR, RQUICKI, NEFA, please write full names, first appearance in article.

Material method

L97: The weight of the animal is not the same as in the abstract (L20), please keep it the same.

The feeding period was only three weeks, and the results were determined whether three weeks had a significant effect. Or need to be fed longer.

2.2. Metabolic challenges: Do you have a stress response to frequent blood collection at different points? Therefore, how to ensure that these external stress factors do not affect the experimental results?

2.3. Tissue biopsies: supplementary RNA extraction methods for tissue samples, and subsequent RT-PCR process.

Results

3.1. Intake and physiological responses

The already published content is mentioned again here. Is it necessary? The author also did not do correlation analysis, just mentioned. If you do not have to, you do not need to explain this section.

L274-278: It is recommended to place it in the table notes or in the material method data analysis.

3.3. Tissue gene expression

Full text genes need italics! Scrutinize.

Discussion

There are too many references in the discussion section, which need to be simplified.

Author Response

L26-28: HOMAIR, RQUICKI, NEFA, please write full names, first appearance in article.

Defined.

L97: The weight of the animal is not the same as in the abstract (L20), please keep it the same.

Corrected to show weight with more significant figures.

The feeding period was only three weeks, and the results were determined whether three weeks had a significant effect. Or need to be fed longer.

According to the data presented in the previous paper (DiGiacomo et al. 2016) physiological and plasma responses to betaine were apparent within the 3 week feeding period.

2.2. Metabolic challenges: Do you have a stress response to frequent blood collection at different points? Therefore, how to ensure that these external stress factors do not affect the experimental results?

Jugular catheters were inserted to attempt to reduce the stress upon the animas, and sheep were acclimatized to the handlers and facilities as much as possible. However, external stressors could impact the experimental results. A line was added to the following sentence (new section in italics) “Chronic indwelling catheters were fitted into the external jugular 2 h prior to the IVGTT (at 0800 h) on day 21 to reduce the stress associated with repeated blood collections.”

2.3. Tissue biopsies: supplementary RNA extraction methods for tissue samples, and subsequent RT-PCR process.

Apologies for this oversight, methods have been added (section 2.4).

3.1. Intake and physiological responses

The already published content is mentioned again here. Is it necessary? The author also did not do correlation analysis, just mentioned. If you do not have to, you do not need to explain this section. L274-278: It is recommended to place it in the table notes or in the material method data analysis.

This section has been truncated and simply cites the published paper. Line 274-278 moved to the data analysis section.

3.3. Tissue gene expression

Full text genes need italics! Scrutinize.

Updated throughout.

There are too many references in the discussion section, which need to be simplified.

Where appropriate some references have been removed and the discussion condensed overall.

Reviewer 3 Report

This paper evaluated the effects of betaine supplementation on ewes experiencing increased temperatures. This is extremely relevant research necessary so producers can implement methods to help combat increasing temperatures. The manuscript is very easy to follow along with, and I genuinely enjoyed reading it. However, some of the methods need further explained and the authors need to be sure to include p-values whenever they say there was no change, an increase, or a decrease.

Abstract:

-No p-values were included in the abstract. Please update.

Line 142: Current sentence reads funny. I would suggest changing to something like: "...collection. Plasma was harvested by centrifugation at ...."

Line 235-238: When discussing mRNA, abbreviations should be italicized. When discussing protein, abbreviations are not italicized. For example, mRNA: HSL, protein: HSL. Please fix here and throughout the manuscript. 

Line 235: The housekeeping gene. Only one housekeeping gene was used. Did the authors test B-actin to ensure that it was not being impacted by the treatments? Additionally, please provide justification for choosing B-actin as a housekeeping gene in skeletal muscle.

No information was provided in the methods on RNA isolation or PCR protocols. This needs to be included. 

Additionally, what was the RNA quality for the samples used? Please include in manuscript.

Some of the figures appear washed out in color. I would suggest fixing throughout the manuscript. 

Line 424: No p-value provided

Line 427-430: No p-value provided. Please double check throughout the manuscript and ensure whenever a statement is made, p-values are provided. 

Author Response

This paper evaluated the effects of betaine supplementation on ewes experiencing increased temperatures. This is extremely relevant research necessary so producers can implement methods to help combat increasing temperatures. The manuscript is very easy to follow along with, and I genuinely enjoyed reading it. However, some of the methods need further explained and the authors need to be sure to include p-values whenever they say there was no change, an increase, or a decrease.

Thank you for this positive feedback.

-No p-values were included in the abstract. Please update.

P-values added.

Line 142: Current sentence reads funny. I would suggest changing to something like: "...collection. Plasma was harvested by centrifugation at ...."

Sentence updated.

Line 235-238: When discussing mRNA, abbreviations should be italicized. When discussing protein, abbreviations are not italicized. For example, mRNA: HSL, protein: HSL. Please fix here and throughout the manuscript. 

Updated throughout.

Line 235: The housekeeping gene. Only one housekeeping gene was used. Did the authors test B-actin to ensure that it was not being impacted by the treatments? Additionally, please provide justification for choosing B-actin as a housekeeping gene in skeletal muscle. No information was provided in the methods on RNA isolation or PCR protocols. This needs to be included. Additionally, what was the RNA quality for the samples used? Please include in manuscript.

Apologies for this oversight, these methods/information has been added (a new section 2.4). B-actin has been routinely published as a housekeeping gene by our group (i.e. Chauhan, et al. 2014; doi:10.2527/jas.2014-8047) and in this experiment statistical analysis was conducted to confirm that B-actin was not impacted by treatments.

Some of the figures appear washed out in color. I would suggest fixing throughout the manuscript. 

Thank you, the figures have been checked and updated.

Line 424: No p-value provided

Updated.

Line 427-430: No p-value provided. Please double check throughout the manuscript and ensure whenever a statement is made, p-values are provided. 

Updated throughout.